# Fast Proteome Identification and Quantification from Data-Dependent Acquisition–Tandem Mass Spectrometry (DDA MS/MS) Using Free Software Tools

**DOI:** 10.3390/mps2010008

**Published:** 2019-01-17

**Authors:** Jesse G. Meyer

**Affiliations:** 1Department of Chemistry, University of Wisconsin—Madison, Madison, WI 53706, USA; jessegmeyer@gmail.com; 2Department of Biomolecular Chemistry, University of Wisconsin—Madison, Madison, WI 53706, USA; 3National Center for Quantitative Biology of Complex Systems, University of Wisconsin—Madison, Madison, WI 53706, USA

**Keywords:** shotgun proteomics, mass spectrometry, protein quantification, peptide quantification, data-dependent acquisition

## Abstract

The identification of nearly all proteins in a biological system using data-dependent acquisition (DDA) tandem mass spectrometry has become routine for organisms with relatively small genomes such as bacteria and yeast. Still, the quantification of the identified proteins may be a complex process and often requires multiple different software packages. In this protocol, I describe a flexible strategy for the identification and label-free quantification of proteins from bottom-up proteomics experiments. This method can be used to quantify all the detectable proteins in any DDA dataset collected with high-resolution precursor scans and may be used to quantify proteome remodeling in response to drug treatment or a gene knockout. Notably, the method is statistically rigorous, uses the latest and fastest freely-available software, and the entire protocol can be completed in a few hours with a small number of data files from the analysis of yeast.

## 1. Introduction

Tandem mass spectrometry is currently the best method for unbiased, high-throughput protein identification [1]. In fact, the entire yeast proteome can be routinely quantified in under one hour [2,3]. Still, the quantification of proteome remodeling can be a slow and difficult process, and many options are available for the multiple steps of the analysis [4,5,6]. The main aim of this protocol is to identify and quantify proteins starting from raw mass spectrometry data. This protocol can be applied to data for any type of biological study, such as studies on diseased and healthy tissues. The analysis is achieved using a combination of the newest software tools to obtain the quantitative results as quickly as possible. All the tools described in this protocol are freely available and adaptable to different types of workflows, such as isotope labeling [7].

There are several protein quantification strategies available to the proteomics researcher, each with its own strengths and weaknesses (see Reference [8] and Figure 2 from Reference [9]). These strategies include: stable isotope labeling with amino acids in cell cultures (SILAC) [10,11], isobaric labelings such as TMT or iTRAQ [12], or label-free quantification. The main differences between these strategies relate to cost, multiplexing, accuracy, and ease of application to human or mouse samples. It must be noted, however, that compared to isotope labeling methods, label-free quantification is extremely sensitive to external factors such as differences in sample preparation, chromatography, and instrument configuration. Still, if samples are processed in parallel with randomization and are analyzed on the same column at a similar period in time, label-free quantification can be more effective in detecting large protein changes than isotope labeling methods. Although this protocol describes an analysis workflow for label-free quantification, the software tools presented herein are also compatible with the analysis of isotope labeling data.

Although there are several software tools and workflows available for protein identification and quantification from DDA tandem mass spectrometry data, the tools and workflow outlined in this protocol have several advantages. For example, compared to MaxQuant, the main two advantages are (1) the speed and (2) the ability to manually interact with the qualitative and quantitative results. The latter is very important for understanding the quality of the quantification. Also, because the described analysis is modular, there are additional options for each step of the analysis. For example, any other database search algorithm could be used for peptide identification, such as MS-GF+ [13]. Also, because Skyline is used for the quantification in this protocol, which extracts signal for each isotope of the peptide precursor mass over its elution time, the peak areas can be output manually with isotope-level granularity and used with any downstream statistical analysis. For example, the peak areas could be output for each sample, filtered for an arbitrary number of top N peptides, and then input into MSstats or mapDIA [14]. Finally, this protocol uses MS-Fragger for peptide identification, which means it can be easily adapted to find unexpected protein modifications [15]. Despite these advantages, this strategy is limited; the multiple technical steps can be more difficult to implement than other options. Still, the benefits of the described strategy far outweigh the limitations, and in the expected results section we highlight one case where this strategy enabled easy verification of a discrepancy between this protocol and MaxQuant’s results.

## 2. Experimental Design

This protocol describes data analysis only, as there are many other examples of protocols for data collection (e.g., Reference [3]). Alternatively, data from a previously-published study can be downloaded from a public repository for re-analysis. Starting with the raw mass spectrometry data, this protocol describes all analysis steps for peptide and protein identification, quantification, and statistical testing. The method uses the graphical user interface (GUI) for MS-Fragger to identify proteins using database searching [15], PeptideProphet and ProteinProphet to refine those identifications [16,17], Skyline to perform quantification [18], and MSstats to perform statistical testing [19]. The tutorial data is from an Orbitrap Fusion mass spectrometer (ThermoFisher Scientific) with high-resolution precursor mass spectra and low-resolution fragment ion spectra, so the specific settings described for the software reflect this. However, to analyze data from another instrument, such as a Q-TOF, the settings can be changed accordingly. These alternative settings are given in the protocol as needed. 

Researchers planning proteomics experiments who wish to use this protocol should collect biological replicates of their controls and the perturbation of interest. The sensitivity of detecting protein changes will depend greatly on the number of replicates collected and the variability of the data. This protocol should yield clear changes when used for the quantification of significant perturbations, such as drug treatments. The tutorial data is from a previous study looking at single-gene knockouts in yeast [20] and is available from massive.ucsd.edu under the accession MSV000083136 (ftp://massive.ucsd.edu/MSV000083136/raw/). Scheme 1 summarizes the experimental design, including the time needed to complete every stage. The entire tutorial process, including software installation, should be completed within 8 h depending on the speed of the computer used, but only a fraction of this time requires user interaction. An advanced scientist who has a 7th generation Intel i7 processor or later, and is familiar with this workflow, can complete the entire process in only 2–3 h, including statistical testing using MSstats. In comparison, analysis of the same data on the same computer using MaxQuant (v1.6.3.3) required ~6 h, not including the statistical analysis.

### 2.1. Materials

Raw mass spectrometry data from the DDA proteomics experiment (tutorial data available from ftp://massive.ucsd.edu/MSV000083136/raw/);Microsoft .NET Framework 4 (https://www.microsoft.com/en-us/download/details.aspx?id=17851);Java runtime environment version 1.8 (https://java.com/en/download/windows-64bit.jsp);MSconvert Software, version 3.0.18282 (http://proteowizard.sourceforge.net/download.html);FragPipe Software v7.1 (https://github.com/chhh/FragPipe/releases/tag/v7.1);MS-Fragger Software, current version (https://bit.ly/2z6dzXa);Philosopher executable Build: 201809241411 (https://github.com/prvst/philosopher/releases/tag/20180924 scroll down and select “philosopher_windows_amd64.exe”);Skyline Software version 4.2 (https://skyline.ms/project/home/software/Skyline/begin.view).

### 2.2. Equipment

A 64-bit computer with Windows 7 or Windows 10 operating system, at least 8 GB of RAM, at least quad-core i5 processor or equivalent, and at least 50 GB of free disk space.

## 3. Procedure

### 3.1. Install Required Software and Setup Directories; Time for Completion: 1 Hour

Follow the instructions on the developer’s websites to install the software programs described under Section 2.1.Setup your directories. Make a new directory on your computer’s C drive called “C:\FragPipe_Skyline” and move the philosopher executable file to this folder. Within that folder make the folder “C:\FragPipe_Skyline\data” and move the .RAW files here.

### 3.2. Identify Peptides Using Database Searching; Time for Completion: 3 Hours

In this section, you will convert the mass spectrometry data files from their vendor-specific format (in this case .RAW) to a readable, open format called .mzXML. You will identify peptides from your data using the most common strategy called database searching, which compares the tandem mass spectra with a database of all possible peptide sequences predicted from the genome sequence. As part of this identification process, you will include fake “decoy” entries in the database that will allow you to assess how often the process is correct (or the false discovery rate, FDR).

#### 3.2.1. Convert Raw Mass Spectrometry Data to mzXML

Navigate to your system folder containing the raw mass spectrometry data.Select your raw data files (.raw files from Thermo instruments, .wiff files from ABsciex instruments).Right click on the selected files and choose “open with MSconvertGUI”.In the options box below the output directory, adjust the settings to output format = “mzXML”, Binary encoding precision = “64-bit”, and check the boxes next to “write index”, “use zlib compression”, and “TPP compatibility”.In the filter box, select the dropdown box, and choose “peak picking”. Do not change the settings that pop up and click “add”. Your window should look like Figure 1.At the bottom right corner, click “start”, and wait for your files to finish converting to mzXML.

#### 3.2.2. Prepare the Organism-Specific Database Using Philosopher

Navigate to the list of UniProt proteomes in your web browser: https://www.uniprot.org/proteomes/.Type the name of your organism into the search box. With the tutorial data, the data is from *Saccharomyces cerevisiae* (UP000002311).Copy the UniProt ID from the column to the left of its name.Open a windows command prompt (click the “start” button on the lower left corner, type “cmd” and hit enter.Navigate to the location of your philosopher executable using the command “cd [full path to folder]” (Figure 3). For this tutorial, we created a file on the C:\drive with the executables, so we use: cd C:\FragPipe_Skyline\.Initialize your philosopher workspace by typing the following (Figure 2):philosopher_windows_amd64.exe workspace –initwhere the first command is the name of your philosopher executable.Download your organism database and add contaminants and decoys by typing (Figure 2):philosopher_windows_amd64.exe database --prefix rev_ --contam --id UP000002311where the last text after “—id” is the uniprot identifier for your organism, which for the tutorial data is *Saccharomyces cerevisiae* (UP000002311). Do not close the command prompt. You will use this again in a subsequent step.

#### 3.2.3. Peptide Identification by Database Search Using FragPipe Interface to MSFragger

Open FragPipe by double-clicking on Fragpipe.bat.The FragPipe window should pop up and prompt you for the locations of the MSFragger.jar and philosopher.exe. Click “browse” to navigate to their locations or click the download buttons for links to their download locations (Figure A1 in Appendix B).Select the second tab “Select LC/MS Files” and add the .mzXML files we created in step 2 by either dragging and dropping them into the large white box, or by clicking “add files” and navigating to their location (Figure A2).Select the third tab, “sequence DB”, and add the FASTA file we created in step 3 by clicking the “browse” button and navigating to its location (Figure A3).Select the fourth tab “MSFragger”, and click the button on the top left “defaults closed search”. Two boxes will pop up asking to confirm. Click “yes” on both boxes.Change the precursor and fragment mass tolerances to values that reflect your instrument performance. For the tutorial data, the precursor tolerance we will use is 10 ppm. Fragmentation spectra were collected at low resolution in the ion trap, so from the dropdown box to the right of “fragment mass tolerance”, set the value to “ABS” and enter 0.35 (Figure A4). These settings are specific to the type of data collection used for the tutorial data and should be adjusted according to the expected accuracy of the data. For TripleTOF (Q-TOF, AB Sciex) data, suitable settings are 30 ppm precursor mass tolerance and 40 ppm fragment mass tolerance.At the top-right of the “options” section, leave the RAM and threads set to 0, and the program will determine these settings for you. You can set these parameters to reflect your computer’s available resources, but this is not required.Leave the remaining tabs with default settings and select the last tab “run”. Set your output file location by clicking the “browse” button, and then click “run” to start the database searches, PeptideProphet, and ProteinProphet analysis. This step will take around 1 h depending on the speed of your computer.Combine the PeptideProphet output files into a single result file using iProphet. In the command prompt from Section 3.2.2, type:philosopher_windows_amd64.exe iprophet data/*.pep.xml.This step will take approximately 1 h depending on the speed of your computer.

### 3.3. Quantify Peptides with Skyline; Time for Completion: 2 Hours

In this section, you will use the Skyline software to create a library of your identified peptides that includes their observed chromatographic retention time, their mass, and their fragmentation pattern from tandem mass spectra. You will create a document in Skyline that contains the peptides you want to quantify, and then import the raw data to quantify the area of the peptide peaks. Skyline is a flexible tool that supports multiple quantitative mass spectrometry workflows, and there are a number of additional tutorials on the Skyline website (https://skyline.ms).
Open Skyline by clicking the windows start button, typing “Skyline”, selecting Skyline, and hitting enter.On the Startup page, click the option in the top middle, “Import DDA Peptide Search”.Skyline will prompt you to save the document. Save the document, and then Skyline will prompt you with the “Import Peptide Search” box. Set the cutoff score to 0.99, and then click “Add Files …” and navigate to your MSFragger output folder. Select the iproph.pep.xml file. Click “Next” and Skyline will start reading the files and building your spectral library.Skyline will then prompt you to extract chromatograms and should find your .mzXML files. If not, browse to add them (Figure A6 in Appendix C).Skyline will prompt you to optionally remove any common prefix from the file names. Click “remove”, and then it will prompt you to add modifications it found in your database search results. Select the modifications you expect and want to use for quantification, in our case N-terminal acetylation, and click “next”.Skyline will prompt you to configure the full-scan settings used for signal extraction. For our tutorial data, set the precursor charges to “2,3,4,5”, and leave the other defaults unchanged (Figure A7). The mass tolerance default of 10ppm here is specific to the type of data collection used for the tutorial data and should be adjusted according to the expected accuracy of the data. For TripleTOF (Q-TOF, AB Sciex) data, change this value to 30 ppm precursor mass tolerance or a value that matches the accuracy of your instrument.Skyline will prompt you for the database used to search for peptides, the enzyme used to digest to proteins, and the number of missed cleavages allowed. Leave the Enzyme as “trypsin”, the missed cleavages as 1 or the value that matches your MS-Fragger search settings, and click “browse” to navigate to the FASTA file created in step 3 (Figure A8). If another protease was used to digest proteins before the mass spectrometry analysis, such as LysC, this can be specified instead of trypsin here. Click “finish” and Skyline will begin adding the proteins that match the identified peptides.Skyline will then prompt you about what proteins you want to keep. You can filter based on the number of proteins identified and whether or not you will allow duplicate peptides. For the tutorial data, keep the default of 1 peptide per protein, and check the box next to “remove duplicate peptides” (Figure A9). This will remove any peptide in the document that matches to multiple proteins. This is important for quantification because if the peptide could come from many proteins, we cannot be sure which protein is the true source and including such ambiguous matches in a protein’s quantification could be misleading. Skyline will then begin extracting the precursor peaks for the identified peptides. You can proceed with the next steps while Skyline continues to import the raw data. The raw data import will take about 1 h depending on the speed of your computer.

### 3.4. Statistical Testing with MSstats: Time for Completion: 1 Hour

Install MSstats within Skyline by going to the “tools” menu > “tool Store”, and then selecting MSstats from the list along the left side and clicking “install”. The installer will also install R, and may take a few minutes.Go to the “settings” menu > “document settings”. Check the boxes next to “condition” and “BioReplicate” and click OK.Click on the “view” menu and select “document grid”. In the document grid popup box, click on the “views” dropdown menu and select “replicates”. Under the “condition” column, select “disease” for the PIM1 replicates, and “healthy” for the WT replicates. Under the BioReplicate column, assign the number of each biological replicate to each sample (Figure A10). Close the Document Grid box.Once the data has finished importing, go to the menu “tools” > “MS stats” > “group comparisons”. Skyline will take a moment to write a report file for input to MSstats. In the popup box “MSstats group comparison”, name the comparison and leave the other settings as default, then click OK (Figure A11). The “immediate window” will pop up and display the status of the process.After the “immediate window” displays “finished”, the MSstats output will appear in the same directory as the Skyline file.Skyline can be used to directly inspect the changes reported by MSstats. To arrange your Skyline workspace for easy data inspection, go to the “view” menu > “arrange graphs” > “tiled”. Also add the peak area comparison window by selecting “view” > “peak areas” > “replicate comparison”. Drag your “peak areas—replicate comparison” window to the bottom of the master Skyline window and drop it over the down arrow that appears to anchor it at the bottom. Your workspace will then appear as shown in Figure 3.

## 4. Expected Results

The procedure presented in this protocol should detect changes in an abundance of individual proteins given that sufficient replicates are collected to achieve enough statistical power. From the example training data provided, this protocol detects 223 protein changes with an adjusted *p*-value <0.05 (Figure 4, Appendix A). As described in the initial publication of this data [20], the protein Isu1p is altered (Figure 5). When using your own data, if there are no changes in your biological comparison, or your data is too noisy either due to variation introduced during sample processing or data collection, then no changes may be detected. The resulting protein changes can be visualized for inspection in Skyline as shown in Figure 5, or the all the protein changes can be visualized together using volcano plots as shown in Figure 4 (the supplemental R script). Another common way to analyze proteomic changes is to use a pathway enrichment analysis, such as enrichr [21]. Figure 6 shows the KEGG pathway enrichment analysis of the 74 proteins that decreased at least 2-fold in PIM1 knockout yeast, which suggests that the metabolism may be altered in this mutant. Interpretation of the proteomic changes discovered with this protocol should be done in the context of the perturbation used as described in the work of Veling et al. [20].

Finally, the results from the data analysis protocol presented here were directly compared with the quantitative results reported by the previous publication of this data where analysis was done using MaxQuant [5]. Skyline quantification and MSstats significance testing were repeated using settings that more closely mimic those used in the previous publication of this data (5 ppm precursor accuracy, 1-minute XIC windows, MSstats analysis without normalization of medians) [20]. The comparison of the quantification produced by these two workflows revealed overall very good agreement (Figure 7a). However, this comparison did reveal one clear outlier protein that was greatly increased according to MaxQuant, but unchanged according to MSstats—QRI7. Highlighting the value of the data analysis workflow presented here, Skyline enabled quick and easy inspection of this protein’s raw quantification data using the find function in Windows (control+F). Skyline quantification of two peptides from QRI7 showed no obvious difference in this protein between the WT and PIM1 knockout groups (Figure 7b). According to a re-analysis of the same data using MaxQuant, the same two peptides were found using both workflows, but manual inspection of the raw data was not easily performed. There are several possible reasons for this discrepancy, such as the wrong peptide peak was integrated, or the peak was only integrated into some samples and assigned intensities of zeros in others. Therefore, in the case of this outlier, we were able to quickly validate that the QRI7 protein is not altered between the compared conditions.

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
