# Peer review of "Fast Proteome Identification and Quantification from Data-Dependent Acquisition–Tandem Mass Spectrometry (DDA MS/MS) Using Free Software Tools"

_mps, 2019, doi:10.3390/mps2010008_

Reviewer 1 Report

The author is trying to write a protocol showing how to identify and quantify DDA proteome data with several free software. However, even each software is useful for targeting one aim in the whole process, the combined use of them all is not very user-friendly or saving time. I am suggesting using one software MaxQuant to replace them all for the same aim. MaxQuant software is capable of doing all the described functions in one step setting, and it can do database search, peptide mapping, protein identification, aa modification identification, and finally do the normalized quantification for protein, peptide, and even for different modification types. And it is free, too.

Here is the website for MaxQuant:

https://www.biochem.mpg.de/5111795/maxquant

Thermo’s commercial software, Proteome Discoverer, is doing similar combined work as MaxQuant, too.

What’s more, for proteome research, label-free quantification is conquered by numerous labeling quantification, i.e. TMT, SILAC, and iTRAQ, etc. The labeling methods can minimize the experimental variance and technique variance, keep treated and untreated sample handling steps as similar as possible, and run all the samples together under the same mass spec condition. So, the results are more reliable. MaxQuant and Proteome Discoverer can handle both labeled and unlabeled proteome works easily with corresponding settings. So, I don’t see the advantages that the author declared in his protocol.

Last but not least, for the parameter settings in the software that the author listed, he used most of them by default. However, different instruments, different LCMS settings of the same instrument, and different experiment aims will require the software to be set accordingly. It should not be one button click to analyze a huge set of proteome data. So, it will not benefit the readers by publishing the protocol. 

Author Response

Reviewer #1:

Comment 1: The author is trying to write a protocol showing how to identify and quantify DDA proteome data with several free software. However, even each software is useful for targeting one aim in the whole process, the combined use of them all is not very user-friendly or saving time. I am suggesting using one software MaxQuant to replace them all for the same aim. MaxQuant software is capable of doing all the described functions in one step setting, and it can do database search, peptide mapping, protein identification, aa modification identification, and finally do the normalized quantification for protein, peptide, and even for different modification types. And it is free, too.

Here is the website for MaxQuant:

https://www.biochem.mpg.de/5111795/maxquant

Thermo’s commercial software, Proteome Discoverer, is doing similar combined work as MaxQuant, too.

Response 1: In the initial manuscript submission, I did not make clear the benefits of this workflow. To clarify the benefits of this have added the following text to the introduction:

“Although there are several software tools and workflows available for protein identification and quantification from data-dependent acquisition tandem mass spectrometry data, the tools and workflow outlined in this protocol has several advantages. For example, compared to MaxQuant, the main two advantages are (1) the speed and (2) the ability to manually interact with the qualitative and quantitative results. The latter is very important for understanding the quality of the quantification. Also, because the described analysis is modular, there are additional options for each step of analysis. For example, any other database search algorithm could be used for peptide identification, such as MS-GF+ [8]. Also, because Skyline is used for quantification in this protocol, which extracts signal for each isotope of the peptide precursor mass over its elution time, the peak areas can be output manually with isotope-level granularity and used with any downstream statistical analysis. Skyline is also a flexible tool that supports multiple quantitative mass spectrometry workflows, and there are a number of additional tutorials on the skyline website. For example, the peak areas could be output for each sample, filtered for an arbitrary number of top N peptides, and then input to MSstats or mapDIA [9].”

And to the experimental design section:

“An advanced scientist who has a 7th generation intel i7 processor or later and is familiar with this workflow can complete the entire process in only 2-3 hours, which is much quicker than the time required by MaxQuant.”

Comment 2: What’s more, for proteome research, label-free quantification is conquered by numerous labeling quantification, i.e. TMT, SILAC, and iTRAQ, etc. The labeling methods can minimize the experimental variance and technique variance, keep treated and untreated sample handling steps as similar as possible, and run all the samples together under the same mass spec condition. So, the results are more reliable. MaxQuant and Proteome Discoverer can handle both labeled and unlabeled proteome works easily with corresponding settings. So, I don’t see the advantages that the author declared in his protocol.

Response 2: Each of the strategies mentioned by the reviewer here have strengths and weaknesses compared to label-free quantification. I have added a paragraph to the introduction:

“There are several protein quantification strategies available to the proteomics researcher, each with its own strengths and weaknesses ([8] and see figure 2 from[9]). These strategies include stable isotope labeling with amino acids in cell culture (SILAC)[10,11], isobaric labeling such as TMT or iTRAQ[12], or label-free quantification. The main differences between these strategies are cost, multiplexing, accuracy, and ease of application to human or mouse samples. This protocol is focused on label-free quantification, which is advantageous due to the low cost and ease of application to human or mouse samples. However, the software and workflow described in this protocol are also compatible with analysis of isotope labeling data.”

Comment 3: Last but not least, for the parameter settings in the software that the author listed, he used most of them by default. However, different instruments, different LCMS settings of the same instrument, and different experiment aims will require the software to be set accordingly. It should not be one button click to analyze a huge set of proteome data. So, it will not benefit the readers by publishing the protocol.

Response 3: I respectfully disagree that readers would not benefit from this protocol. I do agree that there should be some mention of different settings and how to choose those settings in other situations, so I have added detail throughout the protocol, and I have added the following statement to the “Experimental Design” section:

“The tutorial data is from an Orbitrap Fusion mass spectrometer (ThermoFisher Scientific) with high resolution precursor mass spectra and low resolution fragment ion spectra, so the specific settings described for the software reflect this. However, to analyze data from another instrument, such as a Q-TOF, the settings can be changed accordingly. Those alternative settings are given in the protocol as needed.”

Reviewer 2 Report

The authors describes freely available comprehensive software tools to carry out the proteomic identification and quantification. However, following issue need to be addressed:

* Introduction section could be improved to provide sufficient additional background. Need/ advantage of this approach over other approaches need to be included.

* Limitations of this approach should be mentioned.

*Additional examples in the results section would enhance the significance of these software tools.

*This approach talks about label free quantification. Other quantification approaches eg. using isotopically labelled ones need to be explained whether possible using the listed software tools or not.

*A comparison need to be made with other freely available softwares namely, MaxQuant which supports large scale mass spectrometric data sets analysis, supports all main labeling techniques like silac, di-methyl, TMT and iTRAQ as well as label-free quantification.

Author Response

Reviewer #2:

Comment 1:  Introduction section could be improved to provide sufficient additional background. Need/ advantage of this approach over other approaches need to be included.

Response 1: I agree and have added the following text to the introduction:

“Although there are several software tools and workflows available for protein identification and quantification from data-dependent acquisition tandem mass spectrometry data, the tools and workflow outlined in this protocol has several advantages. For example, compared to MaxQuant, the main two advantages are (1) the speed and (2) the ability to manually interact with the qualitative and quantitative results. Also, because the described analysis is modular, there are additional options for each step of analysis. For example, any other database search algorithm could be used for peptide identification, such as MS-GF+ [8]. Also, because Skyline is used for quantification in this protocol, which extracts signal for each isotope of the peptide precursor mass over its elution time, the peak areas can be output manually with isotope-level granularity and used with any downstream statistical analysis. Skyline is also a flexible tool that supports multiple quantitative mass spectrometry workflows, and there are a number of additional tutorials on the skyline website. For example, the peak areas could be output for each sample, filtered for an arbitrary number of top N peptides, and then input to MSstats or mapDIA [9].”

Comment 2: Limitations of this approach should be mentioned.

Response 2: I agree and have added the following text to the introduction:

“Despite these advantages derived from using flexible tools, this strategy is limited in that the multiple technical steps can be more difficult to implement than some other options.”

Comment 3: Additional examples in the results section would enhance the significance of these software tools.

Response 3: I agree and have added the following text to the introduction:

“Another common way to analyze proteomic changes is to use pathway enrichment analysis, such as enrichr[21]. Figure 6 shows the KEGG pathway enrichment analysis of the 74 proteins that decreased at least 2-fold in PIM1 knockout yeast, which suggests that metabolism may be altered in this mutant.”

Comment 4: This approach talks about label free quantification. Other quantification approaches eg. using isotopically labelled ones need to be explained whether possible using the listed software tools or not.

Response 4: I agree and added the following text to the introduction:

“There are several protein quantification strategies available to the proteomics researcher, each with its own strengths and weaknesses ([8] and see figure 2 from[9]). These strategies include stable isotope labeling with amino acids in cell culture (SILAC)[10,11], isobaric labeling such as TMT or iTRAQ[12], or label-free quantification. The main differences between these strategies are cost, multiplexing, accuracy, and ease of application to human or mouse samples. This protocol is focused on label-free quantification, which is advantageous due to the low cost and ease of application to human or mouse samples. However, the software and workflow described in this protocol are also compatible with analysis of isotope labeling data.”

Comment 5: A comparison need to be made with other freely available softwares namely, MaxQuant which supports large scale mass spectrometric data sets analysis, supports all main labeling techniques like silac, di-methyl, TMT and iTRAQ as well as label-free quantification.

Response 5: I added the following statement to the introduction directly comparing this strategy to MaxQuant:

“For example, compared to MaxQuant, the main two advantages are (1) the speed and (2) the ability to manually interact with the qualitative and quantitative results. The latter is very important for understanding the quality of the quantification.”

And to the “experimental design” section:

“An advanced scientist who has a 7th generation intel i7 processor or later and is familiar with this workflow can complete the entire process in only 2-3 hours, which is much quicker than the time required by MaxQuant. ”

Reviewer 3 Report

In this Protocol article, the author provides a brief tutorial that outlines the analysis of shotgun proteomics data using freely available software tools, including MS-Fragger as search engine and Skyline for quantification. The scope of the article is certainly timely, because it describes a multi-step, multi-program workflow that has not been covered in other protocol articles.

I have evaluated this manuscript from the point of view of someone that frequently advises/supervises students and collaborators with little background in proteomics. In its present form the manuscript is somewhat minimalistic in the sense that little additional information beyond the steps/commands required for the execution of the tools is presented. Therefore, one can argue whether a reader without more advanced computational skills or experience in the field will be able to complete all the steps without assistance, especially with his/her own data. The author should keep in mind that software may be updated and functionalities may change on a more or less frequent basis, and if multiple tools are used in succession, as in this protocol, one change is enough to break the pipeline if one blindly follows the guidelines.

So my strong recommendation is that the author expands the article a bit to make it more accessible to non-specialists. This is in line with the journal scope, which states that "... background information to understand the underlying principles, full experimental details, and comparison with available protocols/methods must be provided".

Specific recommendations are given below:

Section 2.1.: Here, the specific versions of the programs used in this protocol should be explicitly listed, screenshots may easily be outdated in newer versions and readers may be confused.

Section 2.2.: No information about the operating system is given here. Later, it is implied that Windows is used. I think it should also be mentioned here which Java version is required.

Are there any other programs/libraries that need to be present on the computer and are not installed by the tools shown? How much disk space is required?

Section 3.1.: The author could suggest a meaningful organization of the data downloads/installations on the hard disk. For example, in section 3.2.2. (line 88), the program is expected to be on C:\

Section 3.3.: Why suggest to use up to 5 missed cleavages here (line 139) when the recommendation during the MS-Fragger search is 1 (Fig. A4)?

Section 3.4.: One could also mention here or elsewhere that there are a number of Skyline tutorials available on the project website.

Figure A4: From what I understand, the suggestion here is to use 30 GB of RAM for the search, what happens if this is not available (the recommendation given earlier is min. 8 GB). This figure should be updated (see also comment above) or explicit recommendations should be added to the text.

In general, terms and concepts that the casual user may not be familiar with should be explained and put into context (e.g. what are the alternatives, what does it do?):

    mzXML (line 65)
    decoys (line 94)
    spectral library (line 127)
    remove duplicate peptides (line 144)
    isotope-level granularity (line 198)

Minor comments:

"... result as quickly as possible" (line 30). In my opinion, this should not be the main motivation for data analysis.

Please spell out GUI (p. 1. line 38).

Author Response

Reviewer #3:

Comment 1: So my strong recommendation is that the author expands the article a bit to make it more accessible to non-specialists. This is in line with the journal scope, which states that "... background information to understand the underlying principles, full experimental details, and comparison with available protocols/methods must be provided".

Response1: Thank you for the positive review of my manuscript and for the suggestions that strengthen it.

Comment 2: Section 2.1.: Here, the specific versions of the programs used in this protocol should be explicitly listed, screenshots may easily be outdated in newer versions and readers may be confused.

Response 2: I agree and have included the software versions in this section.

Comment 3: Section 2.2.: No information about the operating system is given here. Later, it is implied that Windows is used. I think it should also be mentioned here which Java version is required.

Response 3: I agree and have included the software versions in this section. The requirements for Java and .NET were added to section 2.1.

Comment 4: Are there any other programs/libraries that need to be present on the computer and are not installed by the tools shown? How much disk space is required?

Response 4: I believe the addition of Java and .NET cover all the requirements. I have also added the requirement of at least 50 GB of disk space to this section.

Comment 5: Section 3.1.: The author could suggest a meaningful organization of the data downloads/installations on the hard disk. For example, in section 3.2.2. (line 88), the program is expected to be on C:\

Response 5: I agree and have added a step describing setup of the directories.

Comment 6: Section 3.3.: Why suggest to use up to 5 missed cleavages here (line 139) when the recommendation during the MS-Fragger search is 1 (Fig. A4)?

Response 6: Thank you for noticing this detail. I have updated the protocol to use 1 missed cleavage in agreement with the database search. [update figure A4].

Comment 7: Section 3.4.: One could also mention here or elsewhere that there are a number of Skyline tutorials available on the project website.

Response 7: Thank you, this is an important point, and I have added the following statement to the introduction:

“Further, skyline is a flexible tool that supports multiple quantitative mass spectrometry workflows, and there are a number of additional tutorials on the skyline website.”

Comment 8: Figure A4: From what I understand, the suggestion here is to use 30 GB of RAM for the search, what happens if this is not available (the recommendation given earlier is min. 8 GB). This figure should be updated (see also comment above) or explicit recommendations should be added to the text.

Response 8: Thank you for noticing this. I updated the protocol text and the figure to reflect the new text. The additional text reads:

“At the top-right of the “Options” section, leave RAM and Threads set to 0 and the program will determine these settings for you. You can set these parameters to reflect your computer’s available resources but this is not required.”

Comment 9: In general, terms and concepts that the casual user may not be familiar with should be explained and put into context (e.g. what are the alternatives, what does it do?):

 Response 9: I agree I have overlooked explaining these terms. I have added the following descriptions to explain the terms:

 Comment 10:   mzXML (line 65)

                         decoys (line 94)

Response 10: Under section 3.2: “In this section you will convert the mass spectrometry data files from their vendor-specific format (in this case .RAW) to a readable, open format called .mzXML. You will identify peptides from your data using the most common strategy, which compares the tandem mass spectra with a database of all possible peptide sequences predicted from the genome sequence (referred to hereafter as “database searching”). As part of this identification process, you will include fake “decoy” entries in the database that will allow you to assess the how often the process is correct (or the false discovery rate).”

Comment 11: spectral library (line 127)

Response 11: Under section 3.3: “In this section you will use the Skyline software to create a library of your identified peptides that includes their observed chromatographic retention time, their mass, and their fragmentation pattern from tandem mass spectra. You will create a document in skyline that contains the peptides you want to quantify, and then import the raw data to quantify the area of peptide peaks.”

Comment 12: remove duplicate peptides (line 144)

Response 12: In the protocol, I added the following explaination to this step: “This will remove any peptide in the document that matches to multiple proteins, which is important for quantification because if the peptide could come from many proteins, we cannot be sure which protein was the true source, and including such ambiguous matches in a protein’s quantification could be misleading.”

Comment 13: isotope-level granularity (line 198)

Response 13: I rephrased this sentence to read: “Also, because Skyline is used for quantification in this protocol, which extracts signal for each isotope of the peptide precursor mass over its elution time, the peak areas can be output manually with isotope-level granularity and used with any downstream statistical analysis.”

Comment 14: "... result as quickly as possible" (line 30). In my opinion, this should not be the main motivation for data analysis.

Response 14: I agree but realistically we all have many tasks to complete. I believe it is also important to understand how the analysis works, especially for troubleshooting.

I have rephrased this to read: “...faster than alternative tools, such as MaxQuant”

Comment 15: Please spell out GUI (p. 1. line 38).

Response 15: I have spelled-out “graphical user interface (GUI)”.

Round  2

Reviewer 1 Report

The author made lots of changes based on reviewers’ comments. The detailed explanation of the advantages of the software tools did help readers to understand more and get the author’s idea why the software modules were chosen. The description of alternative parameters chosen will also benefit the readers if they want to try different settings when using the tools. However, the author still emphasized too much the advantages of the label-free method and the software modules. Comparing to isotopic labeling method, label-free is too noisy. The different performance of mass spectrometer and the environment condition will affect a huge portion of the results. Two to three-fold changes label-free quantification results may be just due to these external factors. Meanwhile, for yeast whole proteome data with 3-4 technical replicates, the new version of MaxQuant software can also finish all processes and generate results within 2-3 hours. The speed of any search engine is depending on not only the searching algorithm, but also the size of the database and raw files.

Although there were still peccadillos, the proposed methods are a good complementary of MaxQuant or Proteome Discoverer software if users want to separate each step or try different search or quantification algorithms. And the current version is much reader friendly than the previous version. 

Author Response

Comment 1: ....However, the author still emphasized too much the advantages of the label-free method and the software modules. Comparing to isotopic labeling method, label-free is too noisy. The different performance of mass spectrometer and the environment condition will affect a huge portion of the results. Two to three-fold changes label-free quantification results may be just due to these external factors.

Response 1: I agree with the reviewer and have added the following statements to the introduction:

"It must be noted, however, that compared to isotope labeling methods, label-free quantification is extremely sensitive to external factors such as differences in sample preparation, chromatography, and instrument configuration. Still, if samples are processed in parallel with randomization, and analyzed on the same column and period of time, label-free quantification can be more effective in detecting large protein changes than isotope labeling methods. "

Comment 2: Meanwhile, for yeast whole proteome data with 3-4 technical replicates, the new version of MaxQuant software can also finish all processes and generate results within 2-3 hours. The speed of any search engine is depending on not only the searching algorithm, but also the size of the database and raw files.

Response 2: I decided to directly compare the time it takes with MaxQuant and found that only the MaxQuant portion of the analysis, not including the statistical testing in Perseus, takes almost 6 hours. This is about 2x the time it takes can complete the entire process with the proposed workflow. I adjusted the text at the end of section to to reflect this:

" .... complete the entire process in only 2-3 hours including statistical testing by MSstats. In comparison, analysis of the same data on the same computer by MaxQuant (v1.6.3.3) to required ~6 hours not including statistical analysis. "